# Is There Any Benefit to the Use of Antibiotics with Indwelling Catheters after Urologic Surgery in Adults

**DOI:** 10.3390/antibiotics12010156

**Published:** 2023-01-12

**Authors:** Fenizia Maffucci, Chrystal Chang, Jay Simhan, Joshua A. Cohn

**Affiliations:** Department of Urology, Fox Chase Cancer Center, 333 Cottman Ave, Philadelphia, PA 19111, USA

**Keywords:** antibiotic prophylaxis, urinary tract infection, urinary catheter, urology, prostatectomy, urethral diseases, transurethral resection of prostate, catheter-related infections

## Abstract

Antibiotic stewardship in urologic reconstruction is critically important, as many patients will require indwelling catheters for days to weeks following surgery and thus are at risk of both developing catheter-associated urinary tract infections (CAUTI) as well as multi-drug resistant (MDR) uropathogens. Accordingly, limiting antibiotic use, when safe, should help reduce antibiotic resistance and the prevalence of MDR organisms. However, there is significant heterogeneity in how antibiotics are prescribed to patients who need indwelling urethral catheters post-operatively. We performed a literature review to determine if there are benefits in the use of antibiotics for various clinical scenarios that require post-operative indwelling catheters for greater than 24 h. In general, for patients undergoing prostatectomy, transurethral resection of the prostate, and/or urethroplasty, antibiotic administration may be limited without increased risk of CAUTI. However, more work is needed to identify optimal antibiotic regimens for these and alternative urologic procedures, whether certain sub-populations benefit from longer courses of antibiotics, and effective non-antibiotic or non-systemic therapies.

## 1. Introduction

Antibiotic stewardship is a challenging but important consideration in public health and in the care of individual patient. Urologists are routinely tasked with decisions regarding antibiotic selection and duration for both acute infection as well as prophylaxis. Though routinely delivered with the intent of net benefit, there is a great deal of heterogeneity in the paradigms surrounding the use of antibiotic prophylaxis across providers, institutions, and regions at large [1,2,3,4].

Unique and particularly relevant to the urologist are antibiotic practices surrounding indwelling catheterization via any drainage tube that is inserted into the urinary bladder through the urethra or suprapubic region and connected to a closed collection system. There are multiple indications for the use of indwelling catheters, including management of urinary retention, careful monitoring of urine output in critically ill patients, urinary decompression, and support for healing following lower urinary tract procedures. Depending on the indication, indwelling urethral catheters may remain in situ post-operatively for days to weeks at a time. 

Indwelling urethral catheters, though necessary in many post-operative settings, are long recognized as *nidi* for urinary tract infection (UTI). Nevertheless, there is considerable variability in the administration of prophylactic antibiotics to patients with indwelling urethral catheters following urological procedures [1,2,3]. Specifically, certain providers prescribe continuous antibiotics for the duration of catheterization, others prescribe antibiotics for 24–72 h surrounding catheter removal, and some will not prescribe antibiotics at all [1]. 

The rationale for the use of prophylactic antibiotics in the setting of catheterization after lower urinary tract reconstruction is to decrease the likelihood of symptomatic infection and/or to reduce the likelihood of bacteriuria and associated debris, which may result in catheter obstruction. Catheter-associated urinary tract infections (CAUTI) contribute significantly to healthcare costs at an estimated $115 million to $1.82 billion annually [5]. In most cases, post-operative CAUTI results in mild discomfort for the patient. However, more severe consequences of infection can include severe bladder spasms, incontinence, and sepsis. In the most dreaded scenario, a patient may experience poor catheter drainage secondary to an infection, which can compromise complex surgical repairs, resulting in significant morbidity. 

There remain no urologic guidelines that recommend routine utilization of prophylactic antibiotics during a period of prolonged catheterization. Continuous prophylactic antibiotics are not explicitly recommended in the setting of prolonged catheterization due to a lack of evidence that true infections are prevented [6,7,8]. Accordingly, there are knowledge gaps and inconsistencies within the existing literature, societal guidelines, and common practice surrounding the use of antibiotic prophylaxis for post-procedural prolonged indwelling urethral catheterization. It is, therefore, worthwhile to identify if there are circumstances in which antibiotic prophylaxis for catheterization in these settings may be warranted. Should these circumstances exist, then unfavorable infectious outcomes may be prevented with the administration of antibiotics. If not, then antibiotic stewardship may improve by increasing provider confidence in *not* prescribing antibiotics, thus reducing the development of multi-drug resistant (MDR) organisms.

Comprehensive Cochrane reviews from 2012 and 2013 on antibiotic prophylaxis for short-term bladder drainage in adults [9] and on catheter policies for long-term bladder drainage [10] were conflicted regarding the benefit of antibiotic prophylaxis in the setting of postoperative indwelling catheterization, as are existing societal guidelines [6,7,8,11]. Given the lack of consensus or comprehensive review specific to the post urological surgical setting, we explore the use of post-procedural prophylactic antibiotics in the setting of post-surgical indwelling urethral catheters across the spectrum of urologic lower urinary tract procedures. Specifically, we wish to review current guidelines and existing evidence on the impact of antibiotic prophylaxis with post-surgical indwelling catheterization of >24 h on rates of lower urinary tract infection/cystitis, febrile UTI, and catheter obstruction. 

### 1.1. The Association between Indwelling Catheters, Bacteriuria, and Infection 

Urethral catheters are indwelling foreign bodies associated with the colonization of microorganisms (bacteria and fungus), which is temporal in nature [2]. While most patients with short-term indwelling catheters do not acquire colonized bacteria in their urine, virtually all patients will eventually demonstrate bacteria in their urine if catheterization continues for a prolonged interval [8,12,13,14]. Each day that an indwelling catheter is maintained, the risk of bacterial colonization increases by 5% to 10% [15]. Common bacterial and fungal colonizers of urethral catheters include *Escherichia coli*, *Enterococcus spp.*, and *Candida spp*. [7,16].

Catheters are also vulnerable to the development of biofilm, which are matrix-enclosed microbial accretions that adhere to biological or non-biological surfaces [17,18]. One of the more worrying complications of bacterial colonization, particularly in the setting of urologic reconstruction, includes urinary obstruction. Crystalline biofilms, particularly those formed by *Proteus mirabilis*, can aggregate and occlude the lumen of a catheter [18]. With the poor flow of urine caused by a blocked catheter, a patient is vulnerable to the development of severe urinary infection, bacteremia, and breakdown of any surgical repairs of the urinary tract. 

Approximately 75–90% of patients with catheter-associated bacteriuria (CAB) will not develop an inflammatory response or other signs or symptoms to suggest infection [7,19,20]. Although commonly asymptomatic and subclinical, CAB may precede CAUTI and related complications [13,14,21]. Table 1 summarizes the definitions of asymptomatic bacteriuria (ASB), CAB, UTI without a catheter, and CAUTI according to the Infectious Diseases Society of America (ISDA), European Association of Urology (EUA), Centers for Disease Control and Prevention (CDC), and American Urological Association (AUA) [7,8,12,22,23,24,25]. CAUTIs are undoubtedly a significant problem on a global scale, and they have been associated with increased morbidity, mortality, hospital cost, and length of stay for hospitalized patients [19,20,21,26,27]. UTIs are the most common type of healthcare-associated infection, accounting for more than 30% of infections reported by acute care hospitals [26,28]. However, the variability in definitions of UTIs across societies highlights the challenges in identifying infection—both in clinical practice as well as an outcome in research settings. To this end, there are many studies that use the term CAUTI when referring to cases of CAB [7,8], and we would venture that ambiguity also extends to the postoperative setting. Heterogeneity in clinical and research settings presents issues in collating data and assessing the quality of evidence [26].

### 1.2. Societal Recommendations and Guidelines on Antibiotic Prophylaxis in Urology 

By definition, antimicrobial prophylaxis is preventive in nature: it may be considered primary prevention (prevention of initial infection) or secondary prevention (prevention of the recurrence or reactivation of an infection) [29]. Antibiotic prophylaxis poses a double-edged sword. While its systemic use can, in certain situations, limit the catheter colonization of microorganisms and the development of infections, it can also drive the development of MDR organisms [8,12,13,30]. Antibiotic use also carries a risk of adverse side effects and increased medical costs. Thus, a delicate balance must be achieved in order to derive net benefit. Experts in urology and infectious diseases have synthesized available evidence and generated guidelines regarding the use of antimicrobial prophylaxis in the context of indwelling catheterization. Table 2 summarizes relevant guideline recommendations from the AUA, ISDA, EAU, and CDC. Notably absent are specific recommendations regarding the use of continuous antibiotic prophylaxis (CAP) for postoperative indwelling catheters after lower urinary tract reconstruction. 

This is most likely related to a relative lack of evidence and relevance as compared to indwelling catheters in other settings (e.g., temporary catheterization in acutely ill patients or patients undergoing non-urologic surgery). The AUA recommends antibiotic prophylaxis at the time of postoperative catheter removal in certain settings [11], whereas EAU guidelines recommend against antibiotic prophylaxis in this setting given the relative societal risks as compared to the likelihood of individual patient benefit [8].

## 2. Materials and Methods

### 2.1. Aim/Purpose

As outlined previously, the available societal guidelines from the AUA, EAU, IDSA, and CDC do not explicitly recommend the routine use of continuous prophylactic antibiotics to prevent CAB or CAUTI in patients with indwelling urethral catheters. Across these groups, there are conflicting recommendations for or against antibiotic prophylaxis at the time of catheter removal or exchange based largely on patient factors. 

With this review, we aim to explore if there are clinical scenarios in which the use of antibiotics in patients with temporary indwelling urinary catheters remaining longer than 24 h is of benefit following urologic procedures. We then summarize the findings for each clinical scenario and highlight our recommendations (in bolded text). We adhered to standards as outlined by Baethge et al. to optimize review quality and relevance [31].

### 2.2. Review Framework

#### 2.2.1. Types of Studies

We evaluated studies centered on antibiotic prophylaxis for post-operative catheterization in patients who underwent urologic procedures with an anticipated duration of catheterization > 24 h. We included randomized controlled trials, prospective descriptive studies, retrospective quasi-experimental studies, and surveys.

#### 2.2.2. Participants

We included studies with adults (age ≥ 18 years) requiring post-operative indwelling urethral catheterization (regardless of catheter size, design, or material) following urologic procedures, including radical prostatectomy, transurethral resection of the prostate, holmium laser enucleation of the prostate (HoLEP), transurethral resection of bladder tumor, bladder diverticulectomy, partial cystectomy, urethral diverticulectomy, and urethroplasty.

#### 2.2.3. Interventions and Comparisons

Interventions considered were continuous antibiotic prophylaxis for the duration of catheterization, antibiotic prophylaxis at the time of catheter removal only, and no antibiotic prophylaxis. 

#### 2.2.4. Article Search

We performed an electronic search to identify relevant studies from PubMed. We limited our search to full-text articles that were available in the English language. No automation screening tools were used.

The search terms utilized were:

(antibiotic prophylaxis)

AND

(urinary catheter)

AND

(prostatectomy) or (RALP) or (transurethral resection of the prostate) or (TURP) or (holmium laser enucleation of the prostate) or (HoLEP) or (transurethral resection of bladder tumor) or (TURBT) or (bladder diverticulectomy) or (partial cystectomy) or (urethral diverticulectomy) or (urethroplasty).

Authors FM and CC independently assessed all titles and abstracts identified by the search for relevance to the topic. Where there was any possibility that the study might be included, the full paper was obtained and reviewed. Where discordance existed, a consensus was reached by FM, CC, and JC. The date of the most recent search of the register for this review was 11/25/2022. 

#### 2.2.5. Outcomes 

Outcomes of interest included symptomatic lower urinary tract infection/cystitis, fever, sepsis, catheter malfunction, and cultures with MDR bacteria. The presence of these outcomes was based on their study-specific definitions rather than pre-existing criteria set by the authors of this review.

## 3. Results

Figure 1 outlines the identification and exclusion of studies for this literature review (See Figure 1, inspired by Page et al. [32]). In our search of the PubMed database, we identified six studies for robotic prostatectomy (1 prospective descriptive study [33], three retrospective cohort studies [34,35,36], and two prospective randomized controlled trials [37,38]), four studies for transurethral resection of the prostate (4 prospective randomized controlled trials [39,40,41,42]), and six studies for urethroplasty (2 cross-sectional surveys [2,3], three retrospective descriptive studies [43,44,45], and one prospective controlled trial [46]).

In our PubMed search, we did not find relevant articles concerning antibiotic prophylaxis for indwelling urethral catheterization following holmium laser enucleation of the prostate (HoLEP), transurethral resection of bladder tumor, bladder diverticulectomy, partial cystectomy, or urethral diverticulectomy. 

### 3.1. Radical Prostatectomy 

Patients undergoing radical prostatectomy require temporary use of an indwelling urinary catheter after surgery for primary healing of the vesicourethral anastomosis. Typically, the catheter remains in place for six to fourteen days, depending on the surgeon’s preference.

A 2013 study prospectively examined urine cultures obtained from patients who underwent radical prostatectomy immediately prior to catheter removal on postoperative day (POD) 10. All patients received prophylactic antibiotics surrounding catheter removal with oral ciprofloxacin beginning the night before catheter removal and continuing afterward for a total of seven days. Of 334 patients, 83 (25%) had positive cultures with organisms > 1000 CFU/mL, of which 7% were resistant to ciprofloxacin. The authors concluded that a substantial proportion of prostatectomy patients have positive urine cultures at the time of catheter removal despite the administration of prophylactic fluoroquinolone antibiotics. However, outcomes were favorable overall when culture-specific oral antibiotic therapy was initiated. Nevertheless, the benefits of antibiotic administration in the setting of this study were difficult to assess without a control arm [33].

Shin and colleagues compared infectious outcomes after radical prostatectomy between two different antibiotic protocols. Specifically, 153 patients were administered a cephalosporin for fewer than 2 days, and 160 patients received a cephalosporin for more than 2 days. The presence of bacteriuria was examined at the time of catheter removal on POD 14. The researchers reported that the overall incidence of bacteriuria was 51% post-operatively and significantly higher in patients receiving the shorter antibiotic protocol compared to the longer protocol (57% vs. 45%). However, the incidence of fever was not significantly different between the two groups [34].

Pinochet and colleagues retrospectively examined rates of symptomatic UTI in patients who underwent radical prostatectomy by one of two surgeons. One surgeon routinely prescribed a 3-day course of ciprofloxacin prophylaxis starting the day before catheter removal on POD 11; the other surgeon prescribed no antibiotics prior to catheter removal, which was performed on POD 7. Despite a longer catheterization, the group of patients who received antibiotics was observed to develop fewer UTIs in their post-operative course (3.1 vs. 7.3%). Fever was observed in 11 patients in the non-antibiotic group (2.4%); there were no fevers in patients receiving antibiotic prophylaxis. Based on their statistical analysis, the number needed to treat to prevent one UTI was estimated at 24, and to prevent one case of febrile UTI was 91 [35]. 

While these studies suggest there may be an advantage to the routine use of antibiotic regimens of longer duration, other studies have not. A 2017 study examined rates of CAUTI following a change of practice at their institution from a protocol of a prolonged course of prophylactic antibiotics following radical prostatectomy (perioperative cephalosporin and aminoglycoside followed by oral quinolones until catheter removal) to perioperative prophylaxis only. No significant difference in the incidence of CAUTI was noted [36].

Several studies have also examined administering antibiotics at the time of catheter removal only. In 2019, Berrondo and associates performed a prospective randomized controlled trial to evaluate the role of antibiotic prophylaxis prior to urinary catheter removal after radical prostatectomy in preventing UTI. The 167 patients were randomized to receive antibiotic prophylaxis (consisting of ciprofloxacin the evening before catheter removal and the morning of catheter removal) or to receive no antibiotics. Overall, eight (4.8%) patients developed symptomatic UTIs within 6 weeks of urinary catheter removal. No significant difference in the rate of UTI between the control group and the antibiotic prophylaxis group (5.95% vs. 6.02%) was observed [37]. Another randomized control from Ehdaie and colleagues assessed a 3-day course of antibiotics (their historic protocol) to a 1-day course at the time of catheter removal. Over 3 years, a total of 824 patients were randomized to either treatment. The authors found zero UTI (0%) in the 1-day regimen and three UTI (0.7%) in the 3-day regimen and accordingly declared the 1-day regimen to be non-inferior [38].


**When taken together, these studies suggest that among patients after radical prostatectomy, a peri-operative dose of antibiotics only or a peri-operative dose of antibiotics plus a one-day course around catheter removal for patients may provide adequate prophylaxis against UTI while reducing the duration of antibiotics prescribed.**


Although patients undergoing radical prostatectomy undergo lower urinary tract reconstruction, catheter duration is typically less than 1 week, all patients have male anatomy, and patients are considerably less likely to enter surgery with indwelling hardware or recurrent infections than many other patients undergoing lower urinary tract reconstruction for other indications. Therefore, it is not known if these conclusions can be extrapolated to patients undergoing procedures such as bladder diverticulectomy, partial cystectomy, or fistula repair.

### 3.2. Transurethral Resection of the Prostate (TURP) 

Transurethral Resection of the Prostate (TURP) is an endoscopic procedure commonly performed to help relieve bladder outlet obstruction. In general, the literature centers on a comparison of perioperative prophylactic antibiotics versus no perioperative prophylaxis rather than antibiotic prophylaxis, throughout the duration of post-operative indwelling catheterization. 

In 1994, Raz and associates randomized 101 patients undergoing TURP to either receive a prophylactic antibiotic regimen consisting of a single dose of ceftriaxone at the time of surgery and another dose at the time of catheter removal (3 to 4 days post-operatively) or no prophylactic antibiotics. Over the entire 28-day study period, bacteriuria appeared in six ceftriaxone-treated patients and in 20 control patients. Three ceftriaxone-treated patients developed symptoms of UTI requiring additional antibiotics versus 14 control patients. Moreover, the incidence and duration of fever were shorter in the patients treated with ceftriaxone, supporting the routine use of periprocedural and peri-catheter removal antibiotics [39].

A 1996 RCT by Hall and colleagues evaluated the prevention of UTI after TURP following various lengths of prophylaxis with a quinolone antibiotic (fleroxacin). The authors compared the efficacy of (1) a single perioperative oral dose, (2) a single perioperative intravenous (IV) dose, and (3) an initial perioperative IV dose followed by a daily oral dose until removal of the urinary catheter for up to 6 days. Only one patient developed a UTI (single dose IV group), which occurred 22 days postoperatively. There were no instances of urosepsis, nor was there a significant difference in rates of fever between groups. The researchers concluded that a single oral dose of a fluoroquinolone agent is adequate prophylaxis for patients undergoing TURP [40]. More recently, Jayanth and colleagues randomized patients at a single center undergoing TURP to receive a one-day or three-day course of IV amikacin as prophylaxis and evaluated the rate of bacteriuria as the primary outcome. All patients had their catheters removed on POD 3, and a midstream urine culture was obtained the following day. They found no significant difference between groups in rates of bacteriuria and symptomatic UTI up to 3 weeks after surgery. The rates of antibiotic resistance, however, were significantly greater in the group which received 3 days of antibiotics. The authors concluded that a one-day regimen is non-inferior with respect to bacteriuria and symptomatic UTI, with the added advantage of lower rates of antibiotic resistance [41].

Interestingly, a study by Conn and colleagues found no benefit from the use of any prophylactic antibiotics at the time of TURP. The 200 patients were randomized to receive a prophylactic antibiotic regimen consisting of a single dose of cephradine at the time of surgery and another dose at the time of catheter removal (3 to 4 days post-operatively), or to receive no prophylactic antibiotics at all. Between the two groups, there was no significant difference in rates of fever or UTI (defined as urine culture growth of >100,000 CFU of a single organism). The authors concluded that the short-term antibiotic regimen was not of benefit to patients in this setting [42].


**Overall, investigations into the use of antibiotics in patients with indwelling catheters after TURP tend to support minimizing antibiotic use in patients without preoperative infection to a single perioperative dose and possibly a single prophylactic dose at the time of catheter removal.**


These findings are likely to be applicable to other endoscopic outlet reduction procedures, such as water jet ablation, photo vaporization, and laser enucleation. However, with improved hemostasis, catheter duration and manipulation may be minimized, potentially limiting infectious risk.

### 3.3. Urethroplasty 

Given the typically longer indwelling catheter time utilized after urethroplasty, the practice patterns of reconstructive urologists in postoperative antibiotic utilization range widely. McDonald and colleagues administered a survey regarding antimicrobial practice patterns to 34 international members of the Society of Genitourinary Reconstructive Surgeons (GURS) who commonly perform urethroplasty. 18 to 24% of respondents continue intravenous antimicrobials for longer than 24 h post-operatively. 61% administer oral antimicrobials until postoperative catheter removal, which can occur anywhere between 2 and 4 weeks), and the majority give additional antimicrobials at catheter removal [2]. A more recent online survey examining perioperative management of anterior urethroplasty patients was administered to GURS members in 2019, with 142 members responding. The majority (72.2%) of respondents reported continuing oral antimicrobials until catheter removal [3].

Manjunath et al. retrospectively examined close to 400 patients who underwent urethroplasty by a single surgeon from 2000 to 2012. All patients received preoperative antibiotic prophylaxis and postoperative prophylaxis for 30 days or until catheter removal. The investigators identified 102 (25.6%) positive urine cultures (defined as >1000 cfu/mL of an organism) within 30 days of urethroplasty—cultures were collected if there was a concern for UTI (e.g., spasms, fever). There were no significant differences in stricture recurrence (*p* = 0.36) or wound complications (*p* = 0.42) between patients who had a positive and negative urine culture. On adjusted analysis, positive urine cultures (hazard ratio 1.0, 95% confidence interval 0.6–1.8, *p* = 0.88) were not associated with stricture recurrence. The rates of catheter malfunction were not reported [43].

Several studies have assessed the impact of antibiotic duration. Baas and colleagues performed a retrospective review of patients who underwent urethroplasty from September 2017 to March 2020 by a single surgeon, where patients in group 1 (*n* = 60) received extended postoperative antibiotics for 3 weeks until catheter removal, and patients in group 2 (*n* = 60) received antibiotics for 3 days around catheter removal. They defined a UTI as a positive urine culture or reported lower urinary tract symptoms/fevers treated with empiric antibiotics. There was no significant difference in UTI (6.7% vs. 11.7%; *p* = 0.529) or wound infection rates (3.3% vs. 1.7%; *p* = 0.999) between the two groups [44]. Another study by Kim and colleagues also did not appreciate a difference in complications with extended antibiotics after urethroplasty. In a multi-institutional prospective study, 30-day post-operative infectious complications were evaluated in 900 patients undergoing urethroplasty or perineal urethrostomy at one of 11 centers over 2 years. Patients in the first year (cohort A) received a prolonged course of daily oral antibiotics until catheter removal, whereas those in the second year (cohort B) received antibiotics only on the day of catheter removal. They found that the rate of postoperative urinary tract infection and wound infection within 30 days was 5.1% (6.7% for cohort A vs. 3.9% for cohort B, *p* = 0.064) and 3.9% (4.1% for cohort A vs. 3.7% for cohort B, *p* = 0.772), respectively. Given these findings and concerns about the overprescribing of antibiotics, they did not recommend prolonged antibiotic use after urethroplasty [46].

A recent single-center study examined the infectious outcomes associated with the implementation of a post-urethroplasty antimicrobial administration protocol. All 81 patients were treated with intravenous antimicrobial agents until POD 2. Antibiotics were then resumed the day before the urethrogram was performed 2 to 3 weeks postoperatively. Antibiotics were then continued for another 3 to 4 days after the urethrogram. With this protocol, they found a symptomatic UTI rate of 2.5%; however, a significant limitation was that there was no control group for comparison [45]. 

**In the setting of a negative pre-operative urine culture, existing data do not support prolonged use of antibiotics in patients with indwelling catheters who have undergone urethroplasty, nor antibiotics on the day of catheter removal only, although this is common practice.** Our literature review identified considerable variability in practice, likely related to a deficient evidence base. Additional work, preferably randomized prospective studies, is needed to assess for antibiotic protocols that limit symptomatic UTI and catheter malfunction in both men and women undergoing urethral reconstruction without unnecessary doses. A summary of all articles reviewed is presented in Table 3.

## 4. Discussion

Decision-making regarding the use of antibiotics in the context of lower urinary tract reconstruction continues to be based on a combination of limited scientific evidence, personal experience, and inherited dogma, and accordingly, there is considerable heterogeneity in clinical practice. As it pertains to the use of antibiotics, optimizing an individual patient’s surgical outcomes and minimizing morbidity may conflict with the same goals in public health. All reconstructive urologists have received a call from a postoperative patient with an indwelling catheter reporting increased spasms, burning, debris, or gross hematuria and wondered, first and foremost, if this could have been prevented before calling in an empiric prescription for a course of antibiotics. If that urologist routinely prescribes a daily antibiotic with catheters in place, he or she may decide that daily antibiotics make little difference and stop prescribing them. On the other hand, if that urologist does not routinely prescribe antibiotics, he or she may decide to start doing so indiscriminately. Awareness of the growing crisis of MDR organisms appropriately adds to the conflict between and within urologic surgeons.

With this review, we aimed to add confidence and simplicity to clinical decision-making surrounding antibiotics and postoperative catheter management. However, what we found is that high-quality data are relatively lacking for all but the most common procedures (e.g., TURP, radical prostatectomy) and completely absent for others (e.g., bladder diverticulectomy, urethral diverticulectomy). Furthermore, no subpopulations have been identified as a group that would benefit from additional antibiotic prophylaxis. However, this is more a story of “absence of evidence” rather than “evidence of absence.”

Societal guidelines are helpful but ultimately only as good as the data on which they are based, which is, unfortunately, limited. The AUA Best Practice Statement [11] and EAU guidelines [8] may be expected to be the most relevant to antibiotic management following urologic reconstructive surgery. However, neither addresses the scenario of “precious” postoperative catheters whose malfunction could result in failed surgical repairs. Furthermore, AUA and EAU societal recommendations are not in agreement. The AUA recommends antibiotic prophylaxis at the time of catheter removal for patients at high risk of consequences from bacteremia (e.g., immunocompromised, recent joint replacement). On the other hand, the EAU weakly recommends that antibiotic prophylaxis not be administered at the time of catheter removal, balancing societal risks and patient benefits, and strongly recommends that antibiotics *not* be administered with indwelling catheters. The IDSA, whose experts may be most acutely aware of the dangers of the rising prevalence of MDR pathogens, recommend antibiotics to prevent infection associated with the presence, placement, or removal of catheters [7,47]. 

It would be reasonable to extrapolate societal recommendations to urologic reconstruction and recommend against continuous antibiotic administration with indwelling catheters and against any or at least routine use of antibiotics at the time of catheter removal. However, a close review of references on which societal recommendations are based finds that they may come from meta-analyses encompassing a fairly heterogeneous group of non-urologic procedures and even non-urinary drains [8,11]. This speaks to a relative lack of evidence specific to the field of urology and particularly urologic reconstruction. That said, we have found that for TURP, radical prostatectomy, and urethroplasty, the existing albeit limited data support societal recommendations to avoid antibiotics in the typical patient. Unfortunately, we found no randomized urethroplasty studies and no studies at all specifically evaluating antibiotic use in other forms of lower urinary tract reconstruction, such as bladder diverticulectomy or urethral diverticulectomy. 

Importantly, the outcomes of existing studies center around the presence of bacteriuria or the development of symptomatic infection and do not address more surgically relevant outcomes of catheter malfunction and associated morbidity. Prevention of bacteriuria is likely not possible [7,12,14,15,19,20] and, in many patients, irrelevant. Furthermore, symptomatic infection, while uncomfortable, may not impact outcomes sufficiently to justify more widespread use of antibiotics. Urosepsis is an unfortunate but rare complication, but catheter malfunction is not; it is perhaps just harder to identify and its impact more challenging to assess. We would therefore encourage procedure-specific, prospective, randomized studies in reconstructive urology that assess the role of daily antibiotic prophylaxis and peri-catheter antibiotic administration with outcomes that include not only asymptomatic bacteriuria and symptomatic infection but also measures of catheter-associated discomfort, gross hematuria, and catheter malfunction requiring irrigation, exchange, and/or early removal.

What is also apparent is the need for non-antibiotic strategies to prevent postoperative CAUTIs and catheter malfunction. In the setting of chronic and short-term catheterization, these strategies have included antibiotic and non-antibiotic bladder irrigations [48,49,50], non-antibiotic supplements [51,52,53,54], alternative catheter materials and coatings [55,56,57,58,59,60,61,62,63,64], and microbiome alteration [65,66,67]. Table 4 summarizes these alternatives to systemic antimicrobial prophylaxis. Future studies will hopefully add to this armamentarium and identify which are beneficial to a particular subset of patients—and which add cost without sufficient benefit.

Limitations of our manuscript are those inherent to literature reviews, including potential author bias and failure to identify and therefore include studies relevant to the topic. We aimed to mitigate these risks with multi-author identification and review of studies for inclusion and broad search criteria. In addition, this review is limited by the relative quality of the literature, requiring liberal inclusion criteria (i.e., not limiting studies to randomized controlled trials). Nevertheless, we believe this review has value in highlighting both important findings that inform clinical decision-making and areas where deficient study warrants future research.

## 5. Conclusions

Antibiotic stewardship demands limiting the use of antibiotics in patients undergoing urologic procedures who require indwelling catheterization for >24 h. Limited data in patients following TURP, radical prostatectomy, and urethroplasty suggests that, in general, avoidance of continuous prophylaxis does not result in a greater risk of symptomatic infection. A single dose of antibiotics at the time of catheter removal may be warranted in some patients to prevent infection. However, this population has yet to be defined. Opportunities for improvement in future study design include additional outcome variables that improve our understanding of the impact of catheter malfunction on outcomes and the impact of antibiotics on catheter malfunction. Given the concerning rise in the prevalence of MDR pathogens and the association of MDR organisms with systemic antibiotic administration, the development of non-antibiotic strategies is paramount.

## Figures and Tables

**Figure 1 antibiotics-12-00156-f001:**
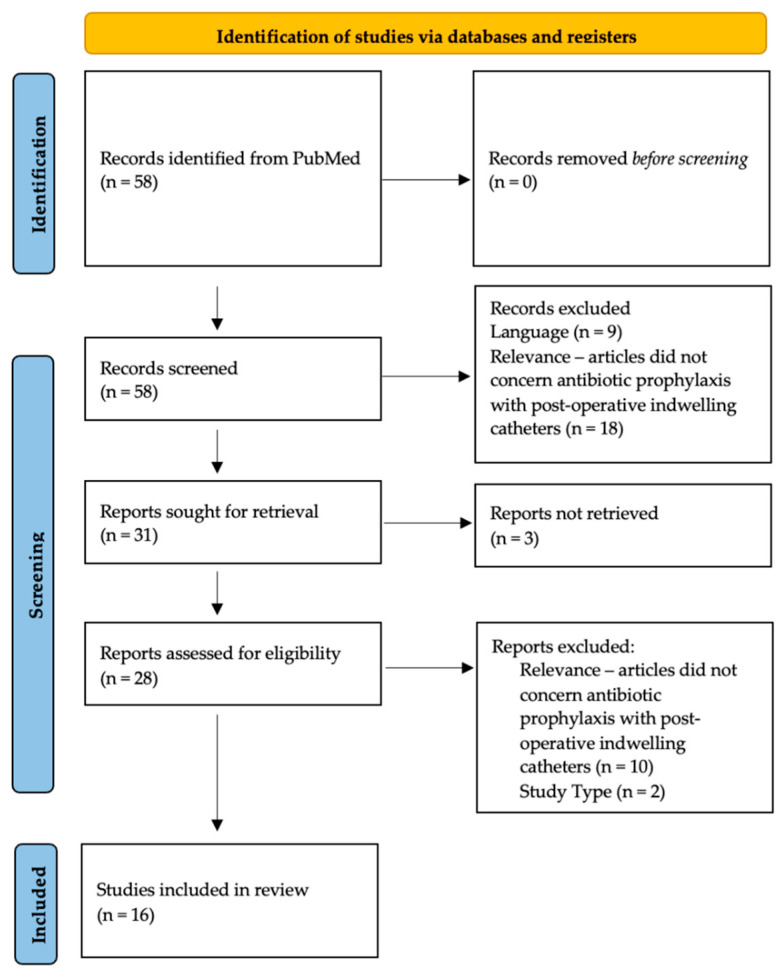
Identification and exclusion of studies for this literature review.

**Table 1 antibiotics-12-00156-t001:** Societal definitions of asymptomatic bacteriuria and urinary tract infections with and without catheters.

Association	Asymptomatic Bacteriuria (ASB)	Catheter-Associated Bacteriuria (CAB)	Urinary Tract Infection (UTI)	Catheter-Associated UTI (CAUTI)
Infectious Diseases Society of America (IDSA)	≥1 species of bacteria growing in voided urine specimen at ≥10^5^ CFU/mL (or ≥10^8^ CFU/L), irrespective of the presence of pyuria, in the absence of signs or symptoms attributable to UTI. For women, two consecutive specimens should be obtained, preferably within two weeks. For men, a single urine specimen is sufficient [12].	≥1 species of bacteria growing in a single urine specimen obtained from indwelling catheter at ≥10^5^ CFU/mL, in the absence of signs or symptoms attributable to UTI [7,12].	Not specifically defined in available guidelines.	Presence of signs or symptoms compatible with UTI with no other identified source of infection along with ≥10^3^ CFU/mL of 1 bacterial species in a single catheter urine specimen or in a midstream voided urine specimen from a patient whose urethral, suprapubic, or condom catheter has been removed within the previous 48 h [7].
European Association of Urology (EUA)	A mid-stream sample of urine showing bacterial growth > 10^5^ CFU/mL in two consecutive voided samples in women or in one single sample in men, in the absence of signs or symptoms attributable to UTI. In a single catheterized sample, bacterial growth may be as low as 10^2^ CFU/mL [8].	Not specifically defined in available guidelines.	The diagnosis of uncomplicated cystitis can be made based on a focused history of lower urinary tract symptoms (dysuria, frequency and urgency) and in the absence of vaginal discharge. CFU not clearly defined [8].	Microbial growth of >10^3^ CFU/mL of ≥1 bacterial species in a single catheter urine specimen or in a mid-stream voided urine specimen from a patient whose urethral, suprapubic, or condom catheter has been removed within the previous 48 h [8].
Center for Disease Control (CDC)	The presence of bacteria in urine. CFU not clearly defined [22].	The presence of bacteria in a urine sample due to bacterial colonization of the urinary tract and/or indwelling urinary catheter. This does not cause symptoms [22].	Patient must meet 1, 2, and 3 below: 1. One of the following is true: • Patient has/had an indwelling urinary catheter, but it has/had not been in place for more than two consecutive days in an inpatient location on the date of event, OR • Patient did not have an indwelling urinary catheter in place on the date of event nor the day before the date of event. 2. Patient has at least one of the following signs or symptoms: • fever (>38 °C) • suprapubic tenderness • costovertebral angle pain or tenderness • urinary frequency • urinary urgency • dysuria 3. Patient has a urine culture with no more than two species of organisms identified, at least one of which is a bacterium of ≥105 CFU/mL [23].	Patient must meet 1, 2, and 3 below: 1. Patient had an indwelling urinary catheter that had been in place for more than 2 consecutive days in an inpatient location on the date of event AND was either: • Present for any portion of the calendar day on the date of event, OR • Removed the day before the date of event. 2. Patient has at least one of the following signs or symptoms: • fever (>38.0 °C) • suprapubic tenderness • costovertebral angle pain or tenderness • urinary urgency • urinary frequency • dysuria 3. Patient has a urine culture with no more than two species of organisms identified, at least one of which is a bacterium of ≥105 CFU/mL [23].
American Urological Association (AUA)	Presence of bacteria in the urine that causes no symptoms [24].	Not specifically defined in available publications.	Symptoms plus 10^5^ CFU/mL of bacteria on a clean catch specimen, or 10^3^ CFU/mL on a catheterized specimen [25].	A UTI that occurs after a catheter has been left in place for 48 h [25].

**Table 2 antibiotics-12-00156-t002:** Summary of guideline recommendations regarding the use of continuous or peri-removal antibiotic prophylaxis with indwelling urinary catheters.

Association	Continuous Antibiotic Prophylaxis during Routine Catheterization	Continuous Antibiotic Prophylaxis during Post-Operative Catheterization	Antibiotic Prophylaxis at the Time of Routine Catheter Removal	Antibiotic Prophylaxis at the Time of Post-Operative Catheter Removal
Infectious Diseases Society of America (IDSA)	Recommend against [7].	Recommend against [7].	Recommend against [7].	Recommend against [7].
European Association of Urology (EUA)	Recommend against [8].	Not specified [8].	Recommend against [8].	Not specified [8].
Center for Disease Control (CDC)	Recommend against [6].	Not specified [6].	Recommend against [6].	Consider in certain settings. Patients with bacteriuria upon catheter removal post-urologic surgery [6].
American Urological Association (AUA)	Not specified [11].	Not specified [11].	Consider in certain settings. Patients at increased risk of urinary infection (e.g., advanced age, immunodeficiency, etc) [11].	Consider in certain settings. Patients at increased risk of urinary infection (e.g., advanced age, immunodeficiency, etc) [11].

**Table 3 antibiotics-12-00156-t003:** Summary of Articles Reviewed.

Study Publication	Study Design	Procedure	Number of Patients Analyzed	Interventions	Outcome Results
Banks et al. 2013 [33]	Prospective Descriptive Single-center	Radical Prostatectomy	334	(1) prophylactic antibiotics (oral ciprofloxacin) for 7 days starting night before catheter removal (*n* = 334)	83 (25%) of patients had positive urine culture results at time of catheter removal, of which 7% were resistant to ciprofloxacin. 2 (0.6%) patients developed symptoms of UTI.
Shin et al. 2017 [34]	Retrospective Cohort Single-center	Radical Prostatectomy	313	(1) prophylactic antibiotics (cephalosporin) administered < 2 days (*n* = 153) (2) prophylactic antibiotics (cephalosporin) administered > 2 days (*n* = 160)	Postoperative bacteriuria at time of catheter removal was significantly higher in group 1 (56.9%) than in group 2 (45%). Fever (>38 °C) was similar (group 1 with 4 fevers, group 2 with 3 fevers).
Pinochet et al. 2010 [35]	Retrospective Cohort Single-center	Radical Prostatectomy	713	(1) prophylactic antibiotics (oral ciprofloxacin) for 3 days starting night before catheter removal (*n* = 261) (2) no antibiotics (*n* = 452)	UTI less common in group 1 (3.1%) than group 2 (7.3%). Fever less common in group 1 (0%) than group 2 (2.4%).
Haifler et al. 2017 [36]	Retrospective Cohort Single-center	Radical Prostatectomy	229	(1) perioperative prophylactic antibiotics, daily until catheter removal (*n* = 60) (2) perioperative antibiotics only (*n* = 129) (Cephalosporin and Aminoglycoside)	CAUTI rate was similar in both groups (8.3 vs. 8.9%, respectively, *p* = 0.89). Logistic regression analysis showed no association between treatment protocol and potential risk for CAUTI.
Berrondo et al. 2019 [37]	Prospective RCT Single-center	Radical Prostatectomy	167	(1) prophylactic antibiotics given prior to urinary catheter removal (2 doses of oral ciprofloxacin, evening prior and morning of) (*n* = 83) (2) no antibiotics (*n* = 84)	8 (4.8%) patients developed symptomatic UTI within 6 weeks of urinary catheter removal. No significant difference in the rate of UTI between the control group and antibiotic prophylaxis group (5.95% vs. 6.02%, *p* = 1). No significant difference in the rates of *C difficile* infection between the control and the antibiotic prophylaxis groups (3.57% vs. 0%, *p* = 0.21).
Ehdaie et al. 2021 [38]	Prospective RCT	Radical Prostatectomy	824	(1) 1 day regimen of prophylactic antibiotics at the time of catheter removal (*n* = 389) (2) 3 day regimen of prophylactic antibiotics at the time of catheter removal (*n* = 435) Ciprofloxacin predominantly used.	0 UTI (0%) in the 1-day regimen and 3 UTI (0.7%) in the 3-day regimen. Declared the 1-day regimen to be non-inferior.
Raz et al. 1994 [39]	Prospective RCT Single-center	TURP	101	(1) antibiotic prophylaxis (ceftriaxone 1 g IV perioperatively and 3–4 days postoperatively at time of catheter removal) (*n* = 51) (2) no antibiotics (*n* = 50)	Fever (> 38.5 C) did not occur in any patient who received antibiotic prophylaxis, but developed in 6 (12%) of the control patients. Duration of fever was also considerably shorter in ceftriaxone treated patients: 10 of 51 had fever more than 2 days vs. 19 of 50 (38%) control patients. Of the six patients in the ceftriaxone group who developed bacteriuria, only three were symptomatic and required antimicrobial therapy. In contrast, 14 of the 20 control patients who were bacteriuric, were symptomatic (*p* < 0.005)
Hall et al. 1996 [40]	Prospective RCT Single-center	TURP	84	(1) single perioperative oral dose of fleroxacin (*n* = 28) (2) single perioperative IV dose of fleroxacin (*n* = 29) (3) initial perioperative IV dose of fleroxacin followed by daily oral fleroxacin until removal of the urinary catheter (up to 6 days) (*n* = 27)	Only one patient developed a UTI (single dose IV group), which occurred 22 days post-operatively. There were no instances of urosepsis nor was there a significant difference in rates of fever between groups.
Jayanth et al. 2021 [41]	Prospective RCT Single-center	TURP	314	(1) perioperative IV amikacin (15 mg/kg) (*n* = 158) (2) IV amikacin for 3 days until catheter removal (*n* = 156)	No significant difference between groups in rates of bacteriuria and symptomatic UTI up to 3 weeks after surgery. The rates of antibiotic resistance were significantly greater in the group which received 3 days of antibiotics.
Conn et al. 1988 [42]	Prospective RCT Single-center	TURP	142	(1) antibiotic prophylaxis (cephradine 1.5 g IV perioperatively and 1 g orally 3–4 days postoperatively at time of catheter removal) (*n* = 74) (2) no antibiotics (*n* = 68)	No significant difference in rates of fever (37.2 C) or UTI (defined as urine culture growth of >100,000 CFU of a single organism).
McDonald et al. 2016 [2]	Cross sectional survey	Urethroplasty	34	27-question survey administered to international members of the Society of Genitourinary Reconstructive Surgeons (GURS)	The majority of reconstruction urologists indicated they would administer prophylactic antibiotics for as long as the catheter is in place. 60.1% selected nitrofurantoin and 21.2% answered fluoroquinolone. The majority stated they give additional antibiotics at the time of catheter removal regardless of a culture 69.7%
Hoare et al. 2021 [3]	Cross sectional survey	Urethroplasty	142	An online survey examining perioperative management of anterior urethroplasty administered to Society of Genitourinary Reconstructive Surgeons (GURS) members.	Postoperatively, oral antimicrobials are routinely administered (70.4%), with most continuing until the urinary catheter is removed (72.2%).
Manjunath et al. 2020 [43]	Retrospective Descriptive Single-center	Urethroplasty	398	All patients received preoperative antibiotic prophylaxis and postoperative prophylaxis for 30 days or until catheter removal. Cultures were collected if there was concern for UTI symptoms.	Identified 102 (25.6%) positive urine cultures (defined as > 1000 CFU/mL of an organism) within 30 days of urethroplasty. There were no significant differences in stricture recurrence (*p* = 0.36) or wound complications (*p* = 0.42) between patients who had a positive and negative urine culture. On multivariate analysis, positive urine cultures (hazard ratio 1.0, 95% confidence interval 0.6–1.8, *p* = 0.88) were not associated with stricture recurrence.
Baas et al. 2021 [44]	Retrospective Descriptive Single-center	Urethroplasty	120	(1) extended postop antibiotic prophylaxis for 3 weeks until catheter removal (*n* = 60) (2) antibiotic for 3 days starting day before catheter removal (*n* = 60) (sulfamethoxazole- trimethoprim 800 mg/ 160 mg twice daily, or cephalexin 500 mg every 8 h)	10 patients had UTIs after urethroplasty. There was no significant difference in UTI (6.7% vs. 11.7%; *p* = 0.529) or wound infection rates (3.3% vs. 1.7%; *p* = 1.000) between the two groups.
Kim et al. 2022 [46]	Prospective Multi-center	Urethroplasty or Perineal urethrostomy	900	(1) prolonged postoperative antibiotics (macrobid 100 mg BID or keflex until catheter removal, plus 2 doses of ciprofloxacin 500 mg or trimethoprim-sulfamethoxazole DS around day of catheter removal) (*n* = 390) (2) antibiotics at time of catheter removal only (2 dosages of ciprofloxacin or trimethoprim-sulfamethoxazole) (*n* = 510)	Rate of post-operative UTI and wound infection within 30 days was 5.1% (6.7% for cohort 1 vs. 3.9% for cohort 2, *p* = 0.064) and 3.9% (4.1% for cohort 1 vs. 3.7% for cohort 2, *p* = 0.772)
Hanasaki et al. 2022 [45]	Retrospective Descriptive Single-center	Urethroplasty	81	All patients were treated with intravenous antimicrobial agents until postoperative day 2. Antibiotics were resumed the day before urethral catheter removal, 2–3 weeks postoperatively. Antibiotics were then continued for another 3 to 4 days after.	Approximately half of the patients had a positive urine culture postoperatively. Wound infections and symptomatic urinary tract infections rates were 3.7% and 2.5%, respectively. No significant correlation was noted with pre- and postoperative positive urine culture. The overall clinical and objective success rates were 96.3% and 79.0%, respectively, and no significant impact of pre- or postoperative positive urine culture was noted.

**Table 4 antibiotics-12-00156-t004:** Alternative approaches to systemic antimicrobial prophylaxis for the prevention of UTI and CAUTI.

Intervention	Summary of Findings	Future Directions
Antibiotic Bladder Irrigations	Patients who perform clean intermittent catheterization who experience recurrent lower UTIs may have reduction in UTI frequency when treated with intravesical Gentamicin [48]. They may also undergo fewer courses of oral antibiotics, and demonstrate less MDR organisms in urine cultures [49]. The side effects are minor [50].	Study utility of antibiotic bladder irrigations in the context of post-procedural catheterization. Study which antibiotic agents are most effective for intravesical irrigations. Compare outcomes of different regimens/frequencies of antibiotic irrigations.
Supplements	Cranberry supplements have been used at UTI prophylaxis in patients who underwent a surgical procedure and required temporary urinary catheterization during the perioperative period. After 4 weeks, the supplement demonstrated a decrease in the occurrence of UTI symptoms, hematuria, and bacteriuria [51]. When studied in patients with neuropathic bladder following spinal cord injury with stable bladder management (indwelling urethral or suprapubic catheter, intermittent catheterization, or reflex voiding with or without a condom drainage device), Methenamine Hippurate (MH) or cranberry supplements did not result in a significantly longer UTI-free period compared to placebo [52]. There are encouraging articles demonstrating the effectiveness and safety of probiotics (with certain strains of lactobacilli) for prophylaxis against uncomplicated UTI [53]. When studied in patients with spinal cord injury, prophylactic Lactobacillus strains did not prevent UTI with more frequency than placebo [54].	Demonstrate a consistent benefit from supplements in the prevention of complex urinary tract infection, in patients with and without catheters. If this can be established, then investigate potential in the post-operative setting.
Catheter selection	There has been inconsistent data regarding benefits of various catheter materials. At present, silicone catheters may be preferred based on equivalent or better outcomes at lower cost. Usage of silicone catheters may result in decreased urethral inflammation and encrustation when compared to other materials [55,56]. Hydrogel-coated catheters aggregate cells and crystals in vitro, leading to catheter blockage [57]. Polytetrafluoroethylene (PTFE), also known as Teflon^®^, coated catheters, inhibit bacterial migration and biofilm formation in vitro [58]. A large RCT compared rates of UTI that occurred following in-strumentation with Teflon-coated silicone catheters vs. nitrofu-razone-coated catheters or silver alloy catheters. They did not find clinically significance difference in rates of UTI between these groups [59]. A trial found PTFE catheters and latex catheters to be associated with increased encrustation compared to those made of silicone [60]. A randomized crossover study found a decreased rate of CAUTI in patients who were randomized to silver-coated catheters ver-sus those with uncoated catheters [61]. Other studies did not demonstrate protective benefit in reduction of CAUTI [62,63]. There has been published preliminary data showing catheters embedded with bioengineered phytomolecules-capped silver nanoparticles may preventing invasion and colonization of uro-pathogens [64].	Explore phytomolecules/nano particles for more effective antimicrobial and anti-encrustation.
Alteration of the urinary microbiome	Patients with decreased proportions of Lactobacillus and an increased number of uropathogens in their pre-operative urinary microbiome have increased risk of post-operative UTI [65]. One technique to alter the microbiome may involve deliberately colonizing the bladder with bacterial interference [66]. Investigators inoculated the bladders of patients with spinal cord injury with a particular strain of *E. Coli*. These patients had less symptomatic UTI compared to controls [67].	Investigate the urinary microbiome, and how it differs in patients who experience recurrent UTI. Explore how urinary microbiome may be altered with catheterization.

## Data Availability

No new data were generated in the writing of this manuscript.

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
