# Peer review of "Is There Any Benefit to the Use of Antibiotics with Indwelling Catheters after Urologic Surgery in Adults"

_antibiotics, 2023, doi:10.3390/antibiotics12010156_

Round 1
Reviewer 1 Report
Very thorough and comprehensive introduction.
The review itself is well done and comprehensive. The article discussion on alternative approaches tends to focus more on neurogenic bladder, and while there is limited data on catheter associated UTI and alternative approaches, there is some data on urinary microbiome and supplements in non neurogenic patients with UTI. Many patients undergoing surgery do not have neuropathic bladders so it would be good to incorporate some data on non-neuropathic patients.
Reviewer 2 Report
Thank you for this thorough narrative review on antibiotic strategies within the peri-operative period for varied urologic procedures.
The introduction was very well put together and complete. I particularly liked the inclusion of guidelines from varied societies.
My major criticism is one of method: Section 5.1 begins Alternative approaches. It was not clear that these were targeted by the search criteria outlined in methods. Indeed, were these included in the manuscript #s in fig 1? Section 4 fails to mention these as well. Ultimately, the search criteria and inclusion criteria for these papers included in section 5 is not established by the manuscript. They also are of tangential to the focused question the article poses, and it is not clear they are needed.
Other specific points:
It is imperative the definition of UTI by each included study is outlined as this may drastically differ among studies and add to heterogenicity and interpretation / generalizability of results . Perhaps a summary table , which could include other pertinent visual cues such as N of each group and outcomes would be of benefit for this summary information.
As the CDC / European guidelines / IDSA all are somewhat against the idea of peri-cath removal antibiotics based on your summary of their position in the introduction, are the papers included that led those guidelines panels to conclude those statements included in your review?
Ultimately the conclusions of your review run somewhat contrarian to 3/4 guideline panel recommendations. This point needs to be discussed at much greater length if your conclusions are to hold. Is new evidence available that was not reviewed by guideline panels? Should conclusions include a re-review by guideline panels?
For discussion I think it important to point out the burden of a UTI in the context of these studies. What % of reviewed papers included a patient that had urosepsis from un-timely treatment? Philosophically are we simply trying to reduce burden on medical offices by covering a medical encounter with abx to meet patient expectations / or fulfill dogma? The NNT to prevent a single febrile UTI was 91 in one study you quoted. Is that number on par with other treatments of mild illness? Is this ethical care given the potential for resistance development in those 91? I'm not sure I have the answers, but perhaps the authors can expand on these points for their readers.
Reviewer 3 Report
Dear Prof. Dr Nicholas Dixon
Editor-in-Chief of the Antibiotics Journal
I am extremely pleased with the opportunity to review the manuscript titled “Is There Any Benefit to the Use of Antibiotics with Indwelling Catheters after Urologic Surgery?”, in which the authors presented a literature review that aimed to “determine if there are benefits in the use of antibiotics for various clinical scenarios that require post-operative indwelling catheters for greater than 24 hours”.
Overall, the writing is clear and sound, congratulations. I believe the manuscript is not structured according to the journal template and required sections (e.g., Introduction, Results, Discussion, Material and Methods, Conclusions). The authors present a subheading 2 (titled Aim/Purpose) that should be deleted as per the journal guidelines.
[Title, abstract, keywords]
- As per international recommendations, please include the study design in the title;
- Please revise the acronym CA-UTI to CAUTI, in line with other international literature on the topic;
- Please consider revising the keywords per MeSH terms to improve future article searchability.
[Background]
- The first sentence of this section is almost identical to the first sentence of the abstract, becoming redundant. Please revise.
- The authors should revise the references according to MDPI’s style.
- Shouldn’t the correct Latin derivation be “nidus”, instead of its plural form (e.g., a source of)?
- Page 2: Please include references that support the sentence “Prophylactic antibiotics are not explicitly recommended in the setting of prolonged catheterization due to lack of evidence that true infections are prevented.”
- Page 2, please consider revising the term “spread” to “development of multi-drug resistant (MDR) organisms”.
- As per international recommendations, the authors should include a paragraph where they discuss the existence of previous literature reviews (narrative or systematic) on this exact topic, including any registered protocols. If there are previous similar reviews, the authors must explain what contributions are expected from their review that was not addressed by previous authors.
- Page 2, subheading 1.1: Please revise the names of specific bacteria and use italics. Likewise, revise the term “species” to spp. (e.g., Candida spp.).
- I fail to understand why the authors included subheadings 1.1, as well as 1.2.1 to 1.2.4, after presenting their objective. It seems like the reader is going back to the initial concepts of the background after being presented with the aim of this article.
- In my opinion, the outlined subheadings do not add much to the article in its present form. While it can be interesting to introduce some detail about the role of biofilms, as well as the different guidance provided by existing international authorities in this field, both sections are not required for the reader to understand the existing gap in the literature. The authors are introducing three more pages of text that could be revised, shortened, and integrated more harmoniously into the previous paragraphs.
[Results]
- It would be a great addition to have a general overview of the 16 included studies that identify the study year, the country in which the study was conducted, the clinical setting, sample sizes, the characteristics of the indwelling catheters (if mentioned), existence of AB protocols, etc.
- Page 6: The authors claim that “We adhered to PRISMA guidelines for our systematic literature review (See Figure 1)”. I have three issues with this statement:
o PRISMA is a checklist, not a guideline, to assist researchers in the reporting of their systematic review process. This means that authors who conduct reviews may choose a specific framework/model/guideline to conduct their review (e.g., Cochrane, Joanna Briggs Institute), and should then report their process and findings as per PRISMA recommendations.
o Figure 1 does not represent the PRISMA recommendations but represents the PRISMA flowchart focused on study retrieval and screening during the different phases of a systematic review.
o Given the narrative nature of their review, I invite the authors to read the SANRA recommendations for quality narrative reviews (https://researchintegrityjournal.biomedcentral.com/articles/10.1186/s41073-019-0064-8).
o The PRISMA flowchart is not correct, more specifically in the first left box. The authors intended to describe how many articles were found in each database or repository used (e.g., 58 articles on PubMed). It is not intended for authors to identify how many databases were searched. Likewise, in the lower right box, the authors should identify why were 10 articles excluded (relevance is vague).
o On page subheadings 4.1 to 4.3, I believe there is some incorrect use of bold text. Please revise.
[Methods]
- It is not clear if the authors followed a specific framework or guidance for the conduction of their narrative review. Likewise, I did not find a review question or mnemonic used.
- Given your inclusion criteria for study participants, should the title and review question and objective also include the term “adults”?
- Page 5, subheading 3.1.: The information presented is not aligned with the subheading (Types of studies). Please specify what study designs were included and/or excluded from this review given the proposed review objective (e.g., randomized controlled trials, quasi-experimental, qualitative studies, expert opinion editorials).
- Page 5, subheading 3.2.: Please revise “examined” to “included”. Were all types of catheters accepted for this review concerning their design (e.g., Coude, Foley), number of lumens and material (e.g., coated catheters, silicone, polyurethane)? This needs to be clarified.
- Did the authors use quotation marks when searching terms that are composed of separate words (e.g, “urinary catheter” instead of urinary catheter)?
- Page 6: The authors state that “We independently assessed all titles and abstracts”. Please clarify how many independent reviewers participated in this process and how did you solve any disagreements between them.
- Page 6: The authors claim that “The date of the most recent search of the register for this review was 11/25/2022”. This sentence should not be included in the outcome subheading. The authors must clarify on which dates the PubMed database search was conducted.
- The authors should explain what definitions, thresholds and/or diagnostic methods were accepted for each of the selected outcomes (e.g., fever, cultures with MDR bacteria).
[Discussion]
- On page 11 the authors claim that “Several investigations tend to support one dose of antibiotics in patients undergoing radical prostatectomy and TURP”. Please include the references that support this sentence.
- The discussion section is not comprehensive and lacks a more incisive overall review of findings from the included 16 studies.
- The authors included a subheading titled “5.1. Alternative approaches”, which is not aligned with the review question or study objectives. Moreover, it seems that the provided alternatives were not withdrawn from the 16 included studies, but based on the authors’ clinical experience. Although interesting, it seems like an odd last-minute addition to the manuscript and diverges from the purpose of a literature review.
- The authors must acknowledge the limitations of their review, starting with the type of review (narrative, not systematic) and outlined inclusion/exclusion criteria (e.g., selection of full text only, use of a single database, simplistic search strategy without controlled vocabulary like MeSH terms).
[Conclusions]
The authors claim that “Future studies will hopefully address any knowledge gaps and perhaps, more importantly, identify effective non-antibiotic options”. While I understand this point, this takeaway message is not enough. I believe the authors can identify why existing studies are of poor quality to support a structured evidence synthesis process (e.g., small sample sizes, lack of diagnostic robustness, study design, and recurrent existing bias), and what needs to be the focus of future research. However, this may be more suitable for a discussion section rather than in the conclusions.
Reviewer 4 Report
The article is interesting and covers the field of antibiotic use with indwelling catheters after urologic strategy. Although comprehensive, I find the introduction too long. I recommend a shorted, more concentrated introduction with highlights on the most important actual level of knowledge in the field.
The aim/purpose of the article is well defined in the specific section but further along the document it is not that well defined. The results section containes a lot of general information ( what do the surgeries consist of, not very relevant data out of the studies included in the article), while discussion section is quite vague. There are two subcategories in discussion section (alternative approaches and supplements) that contained data with little corespondent in the Results or Introduction section.
The references used are a bit old, only 35 out of 67 mentioned references are dated from the last 10 years.
Round 2
Reviewer 2 Report
The article is much better after these revisions. I truly enjoyed reading the discussion and its presentation of the nuance of this difficult topic.
Author Response
Thank you so much for your feedback which helped us to improve this article - and your kind words.
Reviewer 3 Report
Dear Prof. Dr Nicholas Dixon
Editor-in-Chief of the Antibiotics Journal
I am extremely pleased with the opportunity to review the newest version of the manuscript titled “Is There Any Benefit to the Use of Antibiotics with Indwelling Catheters after Urologic Surgery in Adults?: a Review of the Literature”.
Overall, I appreciate the authors' efforts to revise the manuscript according to the provided feedback. I now believe that the methods section is clear and sound.
The writers' use of tables in the Introduction and Discussion sections to lessen the effect of some axilary information is my only criticism. I believe that this approach might not be appropriate outside of the methods and results section, but I will leave this judgment up to the journal's editorial board.
I hope the authors keep up their efforts to synthesize the best available evidence on the use of antibiotics for treating CAUTI.
Author Response
Thank you for your feedback - both on this revised version as well as, of course, on the original version.
We believe that these tables serve as important summaries of important background (when it comes to the introduction) and future directions (when it comes to the discussion) --this provides a framework (thorough but succinct) that defines what we know--and what we hope to know.
Again - thank you so much for your time and incredible effort.